# Trends in renal function in Northern Sweden 1986–2014: data from the seven cross-sectional surveys within the Northern Sweden MONICA study

Julia de Man Lapidoth ![ORCID],[1] Johan Hultdin ![ORCID],[2] P Andreas Jonsson ![ORCID],[1] Maria Eriksson Svensson,[3] Maria Wennberg,[1] Tanja Zeller,[4,5] Stefan Söderberg ![ORCID][1]

[1]Department of Public Health and Clinical Medicine, Umeå University, Umeå, Sweden
[2]Department of Medical Biosciences, Clinical Chemistry, Umeå University, Umeå, Sweden
[3]Renal Medicine, Uppsala University, Uppsala, Sweden
[4]Clinic for General and Interventional Cardiology, University of Hamburg, Hamburg, Germany
[5]German Center of Cardiovascular Research (DZHK), Partner Seite, Hamburg, Germany

**Correspondence to**
Professor Stefan Söderberg;
Stefan.Soderberg@umu.se

## ABSTRACT

**Objective** The prevalence of chronic kidney disease (CKD) is increasing globally, and CKD is closely related to cardiovascular disease (CVD). CKD and CVD share several risk factors (RF), such as diabetes, hypertension, obesity and smoking, and the prevalence of these RF has changed during the last decades, and we aimed to study the effect on renal function over time.

**Design** Repeated cross-sectional population-based studies.

**Setting** The two Northern counties (Norr- and Västerbotten) in Sweden.

**Participants** Within the MONitoring Trends and Determinants of CArdiovascular Disease (MONICA) study, seven surveys were performed between 1986 and 2014, including participants aged 25–64 years (n=10 185).

**Interventions** None.

**Measures** Information on anthropometry, blood pressure and cardiovascular risk factors was collected. Creatinine and cystatin C were analysed in stored blood samples and the estimated glomerular filtration rate (eGFR) calculated using the creatinine-based Lund–Malmö revised and Chronic Kidney Disease Epidemiology Collaboration (eGFR$_{crea}$) equations as well as the cystatin C-based Caucasian, Asian, Paediatric and Adult cohort (CAPA) equation (eGFR$_{cysC}$). Renal function over time was analysed using univariable and multivariable linear regression models.

**Results** Renal function, both eGFR$_{crea}$ and eGFR$_{cysC}$, decreased over time (both p<0.001) and differed between counties and sexes. In a multivariable analysis, study year remained inversely associated with both eGFR$_{crea}$ and eGFR$_{cysC}$ (both p<0.001) after adjustment for classical cardiovascular RF.

**Conclusion** Renal function has deteriorated in Northern Sweden between 1986 and 2014.

## INTRODUCTION

The prevalence of chronic kidney disease (CKD) is increasing globally, and deaths attributed to CKD constituted 4.6% of all deaths in 2017.[1] According to the CaReMe study, 8.3% of the population in Sweden has CKD defined as having a diagnosis, an estimated glomerular filtration rate (eGFR)

## STRENGTHS AND LIMITATIONS OF THIS STUDY

⇒ Study design, selection of participants and measurements consistent over time.
⇒ Population-based study.
⇒ Creatinine and cystatin C measured in the same laboratory during the same period.
⇒ Several risk factors were self-reported.
⇒ Possible residual confounding.

<60 mL/min/1.73 m$^2$ or urine albumin-creatinine ratio ≥30 mg/g.[2] Furthermore, medical treatment due to end-stage kidney disease is costly and constitutes approximately 1% of the total medical expenditures in Sweden.[3]

Decreased renal function is associated with an increased risk for cardiovascular events, including cardiovascular death.[4 5] This is partly due to shared risk factors for CKD and cardiovascular disease (CVD), for example, diabetes, hypertension, obesity and smoking.[6–8] Other possible mechanisms such as systemic inflammation, renal anaemia and endothelial cell dysfunction may contribute.[9] In addition, data suggest that subjects with CVD and CKD get less secondary prevention after a cardiovascular event.[10 11]

In Northern Sweden, the prevalence of hypertension, smoking and hypercholesterolemia has decreased, whereas obesity has increased.[12] In parallel, the use of antihypertensive drugs has increased.[12] Furthermore, low socioeconomic status (SES) is associated with decreased renal function, possibly due to the clustering of other cardiovascular risk factors.[13] Longitudinally, there are, thus, several factors that might have impacted renal function both negatively and positively over time, and the potential net effect is not known.

This study aimed to investigate the development on renal function over 28 years in Northern Sweden, considering the concomitant changes in the prevalence of major cardiovascular risk factors and other factors that may impact renal function over time.

## METHODS

This study is based on seven cross-sectional surveys performed within the framework of the Northern Sweden MONitoring Trends and Determinants of CArdiovascular Disease (MONICA) study.[14] The World Health Organizations' MONICA project was initiated in the mid-80s, and the first survey in the two most northern counties of Sweden (Västerbotten and Norrbotten) was performed in 1986. Altogether, seven population-based cross-sectional surveys using similar methods have been conducted (in 1986, 1990, 1994, 1999, 2004, 2009 and 2014). All surveys were performed during the same months (January–April) to eliminate seasonal variations. The participants have been randomly selected to represent the population and make the different surveys comparable. Two examination teams collected all data using standardised methods to the extent it is possible. Participants were invited if aged between 25 and 64 years in 1986 and 1990 (n=2000 per survey), and 25–74 years from 1994 (n=2500 per survey). Altogether, 11 924 subjects participated in seven surveys with a declining participation rate at each survey (mean 72%, range 81%–61%). The majority of participants were Swedish Caucasians, with a small minority of Finnish and Sami heritage.

### Patient and public involvement

The public or any patient organisations were not involved in study design, analysis of data or in preparing the manuscript.

### Questionnaires

All participants answered questions about diabetes mellitus (yes/no/do not know). Those who answered yes or had diabetes medications were considered as having diabetes. Questions regarding smoking involve active smoking, sporadic smoking, former smoking, former sporadic smoking and non-smoker. The answers were recoded into active smoking (active and sporadic smokers), former smoking (former and former sporadic smokers) and non-smoker. Medications taken regularly were grouped according to Anatomical Therapeutic Chemical Classification-code (ATC-code). Participants who reported treatment with angiotensin-converting enzyme inhibitor (ACEi), angiotensin receptor blocker (ARB) or mineralocorticoid receptor antagonist (MRA) were identified. These treatments were summarised as treatment with a renin-angiotensin-aldosterone system (RAAS) inhibitor.

The self-reported level of attained education was categorised into tertiary education (university), secondary school, and primary school.

### Measurements

Height was measured without shoes using a calibrated scale with a measuring rod attached to the scale or mounted on the wall. The participants wore thin clothing to measure their weight, and a calibrated scale was used (a balance scale 1986–1994, from 1999, an electrical scale was used). The weight was estimated to nearest 0.2 kg. Body mass index was calculated as weight divided by height squared.[15] The waist and hip circumference were measured with a tape measure without clothes between the iliac crest and the lowest rib, over the symphysis bone, and over the part of the hips with the greatest width. With this information, the waist–hip ratio was calculated.[15] After 5 min of rest in the sitting position, blood pressure was measured in the right arm two times at a 3 min interval, and the mean value was calculated. The Hawksley random–zero sphygmomanometer was used between 1986 and 2004, and from 2009 the Omron M7–device was used.[16] Hypertension was defined as systolic blood pressure ≥140 mm Hg, diastolic blood pressure ≥90 mm Hg or the use of antihypertension medication for at least 2 weeks before the survey.

### Blood samples

Blood samples were drawn after fasting for at least 4 hours. Total cholesterol levels were determined at a regional laboratory within 24 hours using an enzymatic method in 1986–1994. From 1999, a dry–chemistry method was used.[17]

In 2015, stored serum samples from all surveys (stored frozen at −80°C) were sent to Hamburg/Germany, as part of the BiomarCaREproject, for further analysis of Cystatin C, Creatinine, N–terminal fragment of the prohormone brain natriuretic peptide (NTproBNP) and high-sensitivity C reactive protein (hsCRP).[18] For approximately 50% of the samples from 1990, EDTA plasma was used instead of serum.

Cystatin C was analysed using a latex immunoturbidimetric assay and creatinine with a Jaffe/kinetic assay (Kinetic Alkaline Picrate), performed on an Architect C8000 system (Abbott, Abbot Park, Illinois). Measurement ranges were for creatinine 17.7–3271 µmol/L with a limit of quantification (LOQ) of 4.4 µmol/L. Creatinine was IDMS (isotope dilution mass spectrometry) traceable (NIST SRM 967). Cystatin C had a measurement range of 0.05–9.6 mg/L, and the LOQ was 0.05 mg/L. hsCRP was analysed with a highly sensitive latex immunoturbidimetric assay. NTproBNP was measured using electro-chemiluminescence sandwich immunoassay ECLIA on an ELECSYS 2010 platform (Roche Diagnostics, Basel, Switzerland).

In addition, in 2009 and 2014 creatinine was measured locally with an IDMS traceable method.

### Statistical analyses

Two different formulas for calculating the creatinine-based eGFR (eGFR$_{crea}$) were used, the Lund–Malmö (LM) revised formula[19] and the Chronic Kidney

Disease Epidemiology Collaboration (CKD–EPI) formula from 2009.[20] To estimate cystatin C-based eGFR (eGFR$_{cysC}$), the Caucasian, Asian, Paediatric and Adult cohort (CAPA) formula[21] was used. For the CKD–EPI calculation, all participants were labelled as Caucasians.

▶ LM revised formula: $e^{X-0.0158\times Age+0.438\times\ln(Age)}$, if female and creatinine <150 $X=2.50+0.0121\times(150-$ creatinine), if female and creatinine $\geq150 X=2.50-0.926\times$Ln (creatinine/150), if male and creatinine <180$X=2.56+0.00926\times(180-$creatinine), if male and creatinine $\geq180 X=2.56-0.926\times$Ln (creatinine/180)

▶ CKD–EPI formula from 2009: $141\times\min\left(\frac{Scr}{\kappa,1}\right)^{\alpha}\times\max\left(\frac{Scr}{\kappa,1}\right)^{-1.209}\times0.993^{Age}\times1.018$ (if female)$\times1.159$(if black), $Scr$=serum creatinine, $\kappa$=0.7 (male) and 0.9 (female), $\alpha=-0.411$ (male) and $-0.329$ (female), min=the minimum of $Scr/\kappa$ or 1 and max=the maximum of $Scr/\kappa$ or 1.

▶ *CAPA* = $130\times\left(cystatin\ C^{-1.069}\right)\times\left(age^{-0.117}\right)-7$

Using the Dubois–Dubois formula,[22] absolute eGFR was also calculated.

$$\frac{eGFR\times(0.007184\times length^{0.725}\times weight^{0.425})}{1.73}$$

Persons aged 25–64.9 years were included in the analysis. Descriptive data included mean values with 95% CIs and medians with IQRs. If necessary, some variables were ln-transformed to improve normality, and geometric means were calculated. Pearson's correlation analyses were performed. To evaluate associations between eGFR and year of survey adjusted for major cardiovascular risk factors, univariable and multivariable linear regression analyses were performed.

Plasma concentrations of creatinine and cystatin C were markedly higher in 2004. Therefore, both results including and excluding data from 2004 are presented.

Most continuous variables were ln-transformed to improve normality. Number of missing data are reported in table 1 and no imputation was done.

To test the robustness of the results, hsCRP and NTproBNP and data from 2004 were excluded from the multivariable model in separate steps, and different formulas for calculation of eGFR were used.

Age and sex were not included in the linear regression or correlation analyses, as these variables are included in the equations of eGFR. However, as a sensitivity analysis, we added age to the multivariable analysis.

For categorical variables with more than two groups, dummy variables were created, including a separate dummy variable for missing data which was omitted from the tables.

The SPSS statistical program V.28 was used (IBM Corporation, New York).

## RESULTS

Baseline data from seven surveys are presented in table 1. The study included 10 185 participants with a mean age (95% CI) of 46 (24–65) years, and 51% were women.

The prevalence of major cardiovascular risk factors changed between 1986 and 2014: total cholesterol, systolic blood pressure and active smoking decreased, whereas BMI increased. hsCRP, NTproBNP and the proportion of the participants using drugs interacting with the RAAS system (ACEi, ARB and MRA) increased (0% to 10.2%). A notable increase in the proportion of participants with higher education, was seen over 28 years.

We found that eGFR$_{crea}$ (both revised LM and the CKD–EPI) and eGFR$_{cysC}$ (CAPA) decreased over time (table 2 and figure 1). In line with this, the percentage of participants with an eGFR below 60 mL/min/1.73 m$^2$ increased. Overall, a more pronounced decrease in renal function (eGFR) was observed over time in Norrbotten county compared with Västerbotten county. Women had higher eGFR$_{cysC}$, but a lower eGFR$_{crea}$ compared with men. The eGFR$_{crea}$ measured locally was lower in 2014 compared with 2009, 84.0 (83.3–84.7) versus 84.8 (84.2–85.4), p=0.03.

In the correlation analysis, study year was associated inversely with eGFR$_{crea}$ (online supplemental table 1), p=<0.001), and with both eGFR$_{crea}$ and eGFR$_{cysC}$ if data from 2004 were excluded (online supplemental table 2), p=<0.001). Age, anthropometric measures, blood pressure, total cholesterol and CRP and NTproBNP were inversely associated with both eGFR$_{crea}$ and eGFR$_{cysC}$ irrespectively if data from 2004 were included or not.

The univariable linear regression analyses displayed the same pattern (online supplemental tables 3,4). Study year associated inversely with eGFR$_{crea}$ (p=<0.001) and with both eGFR$_{crea}$ and eGFR$_{cysC}$ if data from 2004 were excluded (p=<0.001). Furthermore, female sex associated with lower eGFR$_{crea}$ (p=0.001) but with higher eGFR$_{cysC}$ (p=<0.001). Higher attained educational level associated with higher eGFR$_{crea}$ and higher eGFR$_{cysC}$. Active smokers had higher eGFR$_{crea}$ but lower eGFR$_{cysC}$. These results based on the LM and CAPA formulas were almost identical if the CKD–EPI formula was used instead or if absolute eGFR was calculated (data not shown).

In the final multivariable linear regression models, study year remained inversely associated with both eGFR$_{crea}$ and eGFR$_{cysC}$ (table 3, both p=<0.001). This was seen irrespective of including or excluding hsCRP and NTproBNP (data not shown) or data from 2004 (online supplemental table 5). Survey year remained inversely associated with eGFR$_{crea}$ and eGFR$_{cysC}$ if age was included in the multivariable model (data not shown). Lower eGFR$_{crea}$ associated with diabetes, hypertension and higher cholesterol, and with higher CRP and NTproBNP levels, and lower eGFR$_{cysC}$ associated with increasing BMI, hypertension, higher cholesterol, active smoking and with higher CRP and NTproBNP levels. In contrast, active smoking is associated with higher eGFR$_{crea}$. A higher educational level remained positively associated with both eGFR$_{crea}$

**Table 1** Clinical and biochemical characteristics at each evaluated time-point

| | 1986 | 1990 | 1994 | 1999 | 2004 | 2009 | 2014 | Missing |
|---|---|---|---|---|---|---|---|---|
| N | 1604 | 1535 | 1537 | 1432 | 1489 | 1390 | 1198 | – |
| Age | 45.7 (45.1–46.2) | 45.0 (44.4–45.6) | 45.3 (44.7–45.9) | 45.5 (44.9–46.1) | 45.6 (45.0–46.2) | 46.1 (45.5–46.7) | 46.6 (45.9–47.2) | 0 |
| Female sex (%) | 49.4 (46.9–51.8) | 50.8 (48.3–53.3) | 51.3 (48.8–53.8) | 51.5 (48.9–54.1) | 51.8 (49.2–54.3) | 51.7 (49.0–54.3) | 53.0 (50.2–55.8) | 0 |
| BMI (kg/m²)* | 25.0 (24.8–25.2) | 25.1 (24.9–25.3) | 25.5 (25.3–25.7) | 25.9 (25.7–26.1) | 26.3 (26.1–26.5) | 26.3 (26.0–26.5) | 26.5 (26.3–26.8) | 51 |
| Waist (cm)* | 88.5 (88.0–89.1) | 84.5 (83.9–85.1) | 87.0 (86.4–87.7) | 88.5 (87.8–89.1) | 89.9 (89.3–90.6) | 88.0 (87.3–88.7) | 88.0 (87.2–88.7) | 65 |
| Hip (cm)* | 98.0 (97.6–98.4) | 97.6 (97.2–98.0) | 99.4 (99.0–99.8) | 102.8 (102.4–103.2) | 100.5 (100.0–100.9) | 100.8 (100.3–101.3) | 98.10 (97.6–98.6) | 123 |
| SBP (mm Hg)* | 127 (126–128) | 127 (126–128) | 126 (125–128) | 128 (127–129) | 124 (123–125) | 123 (123–124) | 120 (119–121) | 14 |
| DBP (mm Hg)* | 80 (79–80) | 80 (79–80) | 79 (78–79) | 79 (78–79) | 77 (76–77) | 78 (78–79) | 79 (79–80) | 14 |
| Hypertension (%)† | 33.5 (31.2–35.8) | 31.8 (29.4–34.1) | 30.5 (28.2–32.8) | 33.5 (31.0–35.9) | 27.7 (25.5–30.0) | 31.0 (28.6–33.4) | 30.5 (27.9–33.1) | 13 |
| Diabetes mellitus (%)‡ | 3.2 (2.3–4.0) | 2.7 (1.9–3.5) | 1.9 (1.2–2.6) | 2.5 (1.7–3.3) | 2.8 (1.9–3.6) | 3.0 (2.1–3.8) | 3.0 (2.0–4.0) | 0 |
| Active smoking (%) | 31.7 (29.5–34.0) | 31.5 (29.1–33.8) | 29.7 (27.5–32.0) | 22.3 (20.1–24.4) | 21.0 (18.9–23.0) | 17.0 (15.0–19.0) | 12.9 (11.0–14.8) | 40 |
| Former smoking (%) | 29.3 (27.1–31.5) | 27.8 (25.5–30.0) | 26.5 (24.3–28.7) | 28.3 (26.0–30.6) | 28.7 (26.4–31.1) | 28.7 (26.3–31.1) | 29.5 (26.9–32.1) | 40 |
| Cholesterol (mmol/L)* | 6.2 (6.2–6.3) | 6.1 (6.0–6.2) | 5.9 (5.9–6.0) | 5.6 (5.5–5.6) | 5.5 (5.5–5.6) | 5.4 (5.4–5.5) | 5.3 (5.2–5.3) | 49 |
| hsCRP (mg/L)* | 0.85 (0.80–0.90) | 0.92 (0.87–0.98) | 1.12 (1.05–1.18) | 1.15 (1.08–1.23) | 0.85 (0.80–0.90) | 0.98 (0.92–1.04) | 1.10 (1.03–1.17) | 377 |
| NTproBNP (ng/L)* | 38.1 (36.3–40.0) | 38.0 (36.2–40.0) | 43.3 (41.1–45.7) | 41.9 (39.8–44.2) | 31.5 (29.9–33.2) | 38.9 (37.0–41.1) | 42.9 (40.9–45.0) | 1432 |
| Tertiary education (%) | 10.8 (9.3–12.3) | 16.4 (14.5–18.2) | 20.7 (18.7–22.7) | 24.2 (21.9–26.4) | 29.1 (26.8–31.4) | 32.2 (29.7–34.6) | 34.5 (31.8–37.2) | § |
| Secondary school (%) | 22.2 (20.2–24.2) | 22.5 (20.4–24.6) | 29.3 (27.1–31.6) | 35.2 (32.7–37.7) | 39.4 (36.9–41.9) | 41.0 (38.4–43.6) | 48.5 (45.7–51.3) | § |
| Primary school (%) | 62.4 (60.0–64.8) | 60.5 (58.1–63.0) | 49.5 (47.0–52.0) | 40.6 (38.1–43.2) | 30.6 (28.3–33.0) | 25.3 (23.0–27.5) | 15.6 (13.6–17.7) | § |

*Mean (geometric) values with 95% CIs are shown.
†Systolic blood pressure ≥140mm Hg and/or diastolic blood pressure ≥90mm Hg and/or antihypertensive medication.
‡Self-reported and/or on glucose-lowering medication. Primary school=1–6 (9) years and secondary school=7 (10)–12 years.
§Missing data for education level=143.
BMI, body mass index; DBP, diastolic blood pressure; hsCRP, high-sensitivity C reactive protein; NTproBNP, N-terminal fragment of the prohormone brain natriuretic peptide; SBP, systolic blood pressure.

de Man Lapidoth J, et al. BMJ Open 2023;13:e072664. doi:10.1136/bmjopen-2023-072664

**Table 2** Estimated glomerular filtration rate (eGFR) at each time-point

| | 1986 | 1990 | 1994 | 1999 | 2004 | 2009 | 2014 | Missing |
|---|---|---|---|---|---|---|---|---|
| eGFR$_{crea}$ (LM) | | | | | | | | |
| All | 96.3 (95.7–96.9) | 94.9 (94.2–95.6) | 88.3 (87.5–89.1) | 87.6 (86.7–88.5) | 101.5 (100.7–102.4) | 92.7 (91.7–93.6) | 81.6 (80.8–82.5) | 370 |
| eGFR<60 (%) | 0.7 | 0.6 | 2.1 | 3.5 | 0.9 | 3.1 | 5.2 | 370 |
| Men | 96.2 (95.3–97.1) | 95.8 (94.9–96.8) | 89.5 (88.5–90.5) | 87.7 (86.5–89.0) | 101.6 (100.2–102.9) | 93.4 (92.1–94.8) | 82.3 (81.2–83.4) | 173 |
| Women | 96.4 (95.6–97.3) | 94.0 (93.0–95.1) | 87.1 (86.0–88.3) | 87.4 (86.2–88.7) | 101.5 (100.4–102.6) | 91.9 (90.7–93.2) | 81.1 (79.8–82.3) | 197 |
| Norrbotten | 94.9 (94–95.8) | 97.6 (96.6–98.5) | 85.5 (84.4–86.6) | 86.9 (85.9–87.9) | 103.8 (102.5–105.1) | 92.0 (90.6–93.5) | 80.3 (79.1–81.5) | 180 |
| Västerbotten | 97.8 (97–98.7) | 92.5 (91.5–93.5) | 91.4 (90.4–92.4) | 88.3 (86.9–89.8) | 99.3 (98.2–100.6) | 93.4 (92.2–94.6) | 83.1 (81.9–84.3) | 166 |
| eGFR$_{cysC}$ (CAPA) | | | | | | | | |
| All | 100.4 (99.1–101.7) | 92.0 (90.9–93.1) | 85.8 (84.5–87.1) | 88.7 (87.4–90.0) | 121.3 (119.3–123.3) | 97.2 (95.7–98.7) | 84.4 (83.1–85.6) | 380 |
| eGFR<60 (%) | 2.4 | 3.3 | 7.4 | 7.5 | 1.5 | 4.0 | 7.7 | 380 |
| Men | 94.4 (92.8–96.0) | 88.3 (87.0–89.6) | 83.2 (81.5–85.0) | 85.0 (83.3–86.7) | 115.0 (112.3–117.7) | 93.4 (91.5–95.4) | 80.4 (78.9–81.8) | 176 |
| Women | 107.0 (105.1–108.9) | 95.6 (93.9–97.4) | 88.3 (86.4–90.3) | 92.3 (90.4–94.3) | 127.4 (124.4–130.4) | 100.9 (98.8–103.1) | 88.1 (86.2–90.1) | 204 |
| Norrbotten | 98.4 (96.7–100.2) | 94.6 (93.2–96.1) | 80.6 (79.3–82.0) | 88.8 (87.3–90.4) | 124.0 (121.1–127) | 96.9 (94.7–99.2) | 84.0 (82.3–85.7) | 182 |
| Västerbotten | 102.5 (100.7–104.4) | 89.5 (87.9–91.2) | 91.7 (89.4–94.0) | 88.4 (86.3–90.6) | 118.5 (115.7–121.4) | 97.5 (95.7–99.5) | 84.8 (83.0–86.6) | 174 |

Data shown are geometric means with 95% CIs for eGFRcrea and eGFRcysC, stratified for year, sex and county (Norrbotten and Västerbotten).
CAPA, Caucasian, Asian, Paediatric and Adult cohort; crea, creatinine; cysC, Cystatin C; eGFR, estimated glomerular filtration rate (mL/min/1.73 m$^2$); LM, Lund–Malmö revised.

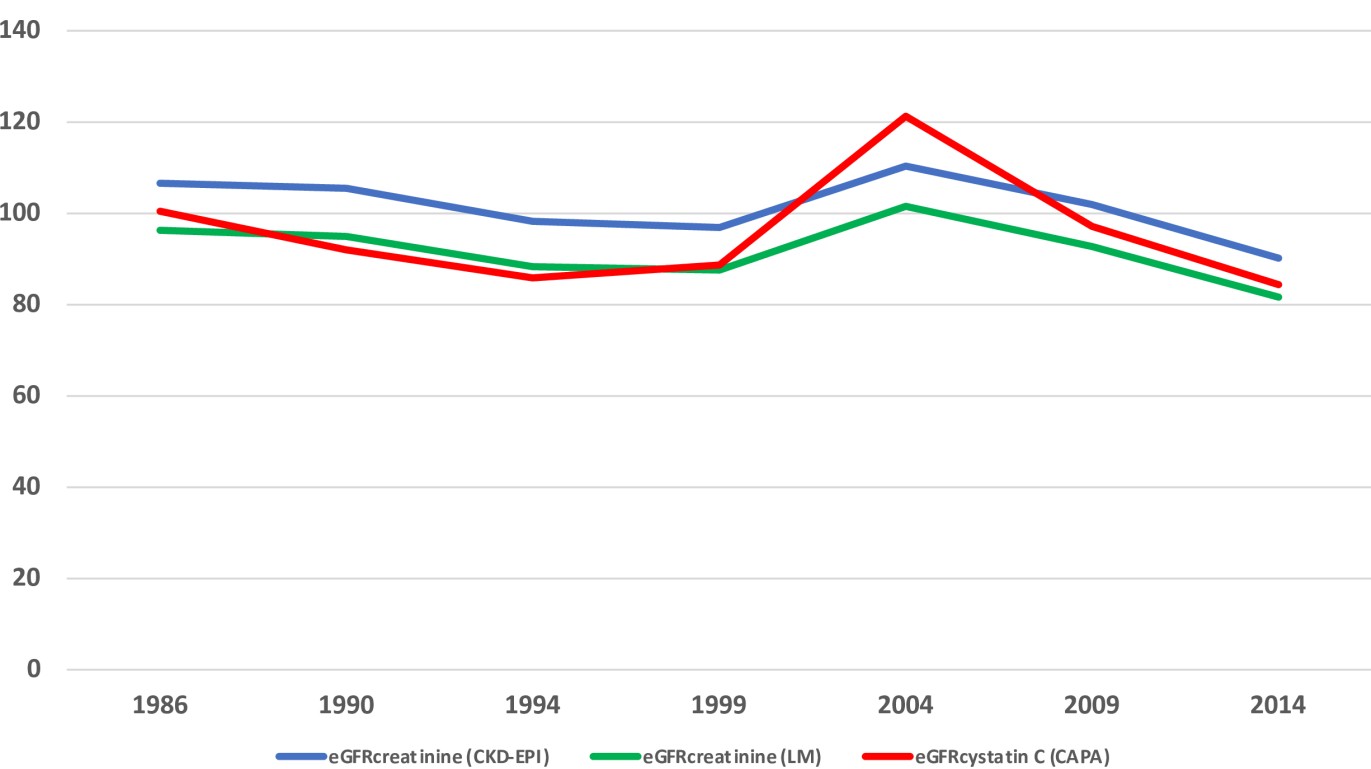

**Figure 1** Renal function in Northern Sweden between 1986 and 2014 expressed as estimated glomerular filtration rate (eGFR) (geometric mean) based on the Lund–Malmö (LM) revised formula and the Chronic Kidney Disease Epidemiology Collaboration (CKD–EPI) formula based on creatinine, and the Caucasian, Asian, Paediatric, and Adult cohort (CAPA) based on Cystatin C.

and $eGFR_{cysC}$, whereas county did not associate with eGFR after adjustments. Using the CKD–EPI equation gave almost identical results (data not shown).

Study year remained inversely associated with $eGFR_{crea}$ and $eGFR_{cysC}$ if hypertension was replaced with systolic and diastolic blood pressure, if the use of ACEi, ARB and

**Table 3** Multivariable linear regression analysis between two different estimated glomerular filtration rates (eGFR) and clinical and biochemical variables including survey year and county

|  | eGFR$_{crea}$ | | eGFR$_{cysC}$ | |
|---|---|---|---|---|
|  | **Unstandardised ß** | **Standardised ß** | **Unstandardised ß** | **Standardised ß** |
| Survey | −0.004*** | −0.187*** | −0.001*** | −0.037*** |
| BMI (kg/m²) | 0.019 | 0.017 | −0.141*** | −0.075*** |
| Hypertension (Y/N)† | −0.013*** | −0.04*** | −0.017** | −0.03** |
| Diabetes mellitus (Y/N)‡ | −0.051*** | −0.046*** | −0.022 | −0.012 |
| Active smoker (vs never smoker) | 0.026*** | 0.061*** | −0.020** | −0.029** |
| Former smoker (vs never smoker) | −0.004 | −0.010 | 0.001 | 0.002 |
| Cholesterol (mmol/L) | −0.118*** | −0.142*** | −0.128*** | −0.091*** |
| hsCRP (mg/L) | −0.023*** | −0.15*** | −0.064*** | −0.246*** |
| NTproBNP (ng/L) | −0.042*** | −0.22*** | −0.056*** | −0.174*** |
| Tertiary education (vs primary school) | 0.009 | 0.022 | 0.046*** | 0.065*** |
| Secondary school (vs primary school) | 0.036*** | 0.094*** | 0.056*** | 0.088*** |
| County (1=Västerbotten, 2=Norrbotten) | −0.005 | −0.014 | 0.004 | 0.006 |

Unstandardised and standardised ß-values are shown, and eGFRcrea and eGFRcysC, respectively, are dependent variables in separate multivariable analysis. eGFRcrea and eGFRcysC according to the LM and CAPA formulas, respectively, and were Ln-transformed.
*p<0.05, **p<0.01, ***p<0.001
†Systolic blood pressure ≥140 mm Hg and/or diastolic blood pressure ≥90 mm Hg, and/or antihypertensive medication.
‡ Self–reported and/or use of glucose lowering medication.
BMI, body mass index; DBP, diastolic blood pressure; hsCRP, high sensitivity C reactive protein; NTproBNP, N-terminal fragment of the prohormone brain natriuretic peptide; SBP, systolic blood pressure.

de Man Lapidoth J, *et al*. *BMJ Open* 2023;**13**:e072664. doi:10.1136/bmjopen-2023-072664

MRA was added, or if the waist and hip circumferences were added (data not shown). High blood pressure and the use of RAAS blocking agents (ACEi, ARB and MRA) were all independently associated with lower $eGFR_{crea}$ and $eGFR_{cysC}$ (data not shown).

## DISCUSSION

In this study, we found that renal function assessed as $eGFR_{crea}$ and $eGFR_{cysC}$ has decreased over 28 years in Northern Sweden. In parallel, the proportion of persons with an eGFR below $60\,mL/min/1.73\,m^2$ increased. This was also seen after adjustment for concomitant changes in the prevalence of major cardiovascular risk factors and other known cardiovascular risk factors that may impact renal function over time. Notably, the renal function differed between the studied counties, possibly due to varying cardiovascular risk profiles, as the difference did not remain after adjustments. Possibly, the ongoing primary–care intervention programme in Västerbotten may have had an impact.[23] We also found that women have higher $eGFR_{cysC}$ but lower $eGFR_{crea}$ than men, which is not in line with earlier studies showing that male gender is usually associated with a lower $eGFR_{crea}$.[24]

These results are in line with increasing prevalence of end-stage renal disease between 1990 and 2021 in Sweden.[25] The incidence is also increasing but not as evident as the prevalence, probably due to better care and longer survival. In contrast, the prevalence of CKD not requiring replacement therapy is largely unknown in Sweden as these alterations are not uniformly registered. The CaReMe-study showed that approximately 8.3% of the Swedish population has CKD.[2]

To our knowledge, a similar study of renal function with repeated measurements of both creatinine and cystatin C over a long time has not been done previously. Several studies have focused on changes in renal function in the same cohort over time, showing deteriorating renal function with increasing age.[24] Two previous population-based studies in Scandinavia have followed renal function over time.[26 27] They were mainly based on creatinine measurements and covered shorter periods of time. In Norway, the prevalence of CKD was stable for more than a decade while the eGFR decreased in Finland over 5 years.

We adjusted for several factors to try to explain the deteriorating renal function. However, the decline in eGFR remained an independent finding in the multivariable analysis. This finding may be worrying as reduced renal function is associated with an increased mortality risk.[28] Hypertension and smoking contribute to a faster decline in renal function,[7] and the use of antihypertensive medications has increased in northern Sweden,[29] including RAAS inhibitors ACEi, ARB and MRA. These medications are known to initially decrease the eGFR due to a reduction of glomerular hyperfiltration[30] and could partly explain an observed decrease in renal function over time. The observed inverse relationship between the use of RAAS–inhibitors could also be due to colinearity

with the conditions (ie., hypertension and renal protection) that they are used for. However, the long-term treatment with these medications preserves renal function.[31]

According to a previous study on this population, the reported percentage of energy intake from protein increased slightly from 14% to 16% from 1994 until 2014. It is unlikely that this small increase had any large impact on the finding of increasing creatinine.[32] In addition, increasing creatinine and a lower estimated eGFR are further supported by the finding of decreasing renal function when using cystatin C to estimate renal function.

Another potential explanation could be increased exposure to environmental pollutants,[33] but previous studies have shown a decrease in lead in the population and a stable amount of cadmium.[34] However, there are numerous other known and unknown pollutants that could potentially affect renal function.

Northern Sweden also has a high incidence and prevalence of the viral infection nephropathia epidemica (Puumala virus),[35] which is a haemorrhagic fever with a renal syndrome. The long-term effects are still unknown, and earlier studies have shown contradictory results if the infection causes irreversible long-term kidney damage.[36 37]

Smoking affected $eGFR_{crea}$ and $eGFR_{cysC}$ differentially, which has been presented earlier.[38] It has been suggested that smoking not only causes glomerular hyperfiltration[39] affecting creatinine more but also that sarcopenia may contribute to lower creatinine values in smokers. Possibly, smoking per se may increase cystatin C levels,[40] which is supported by studies showing a decrease in cystatin C following smoking cessation.[41]

This study shows that higher education is associated with better renal function, which is in line with previous studies.[42] Lower SES is associated with a higher incidence of CKD,[43] and a higher risk of chronic renal failure.[44] Several indicators for SES have been used, such as educational attainment, household, or individual income or geocoordinates. However, similar results related to renal function have been shown irrespective of how SES has been defined.[42]

### Strengths and limitations

The strength of this study is that renal function is measured using two different endogenous markers, creatinine and cystatin C, which are affected by different confounding factors[45 46] and both endogenous markers show the same trend of decreasing renal function. An additional strength is that data are collected from a randomly selected, large cohort, representing the general population at the time of each survey, which makes the results applicable to the general population. One additional strength is that all creatinine and cystatin C analyses for the different years were performed within the same period, at the same laboratory, using the same methods.

There are several limitations as well. Some of the blood samples have been frozen for a long period, the earliest from 1986, which could affect the results of the biochemical analyses. Prolonged storage time could result in the

degradation of biomarkers and sublimation of water from the frozen samples altering concentrations of the biomarkers. Still, the finding of a reduction of both eGFR-$_{crea}$ and eGFR$_{cysC}$ over time suggests that there is an actual reduction in renal function (eGFR). Furthermore, it should be emphasised that the samples have been stored in –80°C from sampling to analysis without thawing. The method to analyse creatinine, the Jaffe/kinetic assay, is not specific and other compounds (eg, haemoglobin F, certain proteins, glucose and ketone bodies) produce Jaffe-like chromogens (also known as pseudocreatinines or non-creatinine chromogens). To adjust for this potential overestimation, compensated analyses can be used. Since all blood samples are analysed using the same method, it is possible to use this method to investigate renal function over time in the population. However, we saw variations in levels of biomarkers related to survey year, the 2004 results in particular. We, therefore, repeated all analyses after excluding 2004 data, which gave similar results. Quality indicates have been scrutinised, but no obvious reason has been found for this discrepancy.

Other limitations lay in that some of the data included in this study are self-reported (diabetes, smoking, medication), which makes recall bias impossible to eliminate. The participation rate has been very good in the MONICA surveys (72%), which is high compared with other population-based surveys internationally. There is always a selection in this kind of studies, and healthier and older individuals are more likely to participate. Since eGFR is an estimation of renal function and not a direct measurement, it is possible that these results are not exclusively based on a decreasing renal function but could also be affected by changes in living conditions and measurements. One known risk factor is alcohol intake.[47] This is not included in this study due to the lack of reliability of self-reported alcohol intake but a previous study on dietary trends in this population found an increase in reported energy from alcohol over time.[32] We also lack information about medications bought over the counter such as non-steroidal anti-inflammatory drug that may impact renal function both short term and long term.[48] Finally, the results are based on a predominately Caucasian population in Northern Scandinavia, and the generalisability to other ethnicities and contexts is unknown.

## Conclusion

In this study, we found that estimated renal function has deteriorated in Northern Sweden independently of changes in cardiovascular risk factors and SES.

**Correction notice** This article has been corrected since it was first published. Table 3 has been updated.

**Acknowledgements** The authors thank the MONICA project, scientific leaders, and participants through the years, as well as the Department of Biobank Research at Umeå University (https://www.umu.se/en/biobank–research–unit/) for providing data and blood samples. The authors also want to thank the MORGAM/BiomarCaRe team in Hamburg and Helsinki for the extensive analyses of biomarkers. A special thanks to Professor Kari Kuulasma in Helsinki, Finland and to Dr Stefan Blankenberg in Hamburg, Germany.

**Contributors** SS is principal investigator for the Northern Sweden MONICA study and is responsible for the repeated population-based surveys and the collection of data and samples. SS initiated this study and drafted the main hypothesis, and is the guarantor for the study, the analysis and for the desicion to publish. TZ is responsible for the analyses of Creatinine and Cystatin C and related qualitative checks. JDL made the primary analyses and drafted the manuscript. JH, PAJ, MKS and MW have together with JDL conducted literature search and prepared the manuscript. JdML, SS, JH, PAJ, MKS, MW and TZ have all been responsible for carefully interpretation of results, contribution to and revision of the manuscript, and they have read and approved the final version of the manuscript, as well as agreed to be accountable for all aspects of the work. SS and JdML have full access to the dataset.

**Funding** The Northern Sweden MONICA study was funded by the county councils in Norr– and Västerbotten, and by Umeå University. The BiomarCaRE Project was funded by the European Union Seventh Framework Programme (FP7/2007–2013) under grant agreement number HEALTH–F2–2011–278913. JdML was supported by ALF from the county council of Västerbotten.

**Competing interests** MKS reports no conflict of interest or competing interests to declare related to this project. Provided expertise in ad boards and given lectures for Amgen, AstraZeneca, Boehringer Ingelheim and GSK. Been a clinical trialist and scientific collaborations with Bayer, MSD, Novo Nordisk, AstraZeneca and Vifor Pharma. SS reports speaker honoraria from Actelion Ltd. All others declare no disclosures.

**Patient and public involvement** Patients and/or the public were not involved in the design, or conduct, or reporting, or dissemination plans of this research.

**Patient consent for publication** Not applicable.

**Ethics approval** This study involves human participants and was approved by Umeå University Ethics Committee: For surveys 2013/97–31 and for biochemical analyses 2012–280–32M. Participants gave informed consent to participate in the study before taking part.

**Provenance and peer review** Not commissioned; externally peer reviewed.

**Data availability statement** Data are available upon reasonable request. Data from the Northern MONICA surveys are not freely available according to GDPR due to the presence of individual data. However, pseudonymised data can be shared after ethical evaluation and formal request (https://www.umu.se/en/biobankresearch–unit/ research/access–to–samples–anddata/access–to–nsdd/).

**ORCID iDs**
Julia de Man Lapidoth http://orcid.org/0000-0001-7592-2526
Johan Hultdin http://orcid.org/0000-0002-9599-0961
P Andreas Jonsson http://orcid.org/0000-0002-1938-0707
Stefan Söderberg http://orcid.org/0000-0001-9225-1306

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
