## [Reviewer comments · BMJ Open]

ARTICLE DETAILS

TITLE (PROVISIONAL)	Trends in renal function in Northern Sweden 1986 – 2014: data from the seven cross-sectional surveys within the Northern Sweden MONICA study
AUTHORS	de Man Lapidoth, Julia; Hultdin, Johan; Jonsson, P. Andreas; Eriksson Svensson, Maria; Wennberg, Maria; Zeller, Tanja; Söderberg, Stefan

VERSION 1 – REVIEW

REVIEWER	Taal, Maarten Royal Derby Hospital, Renal Medicine
REVIEW RETURNED	27-Feb-2023

GENERAL COMMENTS	This analysis investigated trends in estimated GFR values in a total of 10,185 participants in the MONICA study in Sweden from 1986-2014. The primary finding was a reduction in mean eGFR over time which remained significant after adjustment for multiple risk factors. Comments: 1. The relevance of the SCREAM cohort is not clear.2. The introduction discussed the cost of care for people with CKD and ESRD in the US but it may be more relevant to discuss the cost in Sweden.3. Age inclusion changed from 25-64 to 25-74 from 1994. Age was identified as an important determinant of eGFR but not included in the multivariable model because age is a variable in GFR estimating equations. This would only be justified if significant collinearity was observed between eGFR and age. Table 1 reports r-values of 0.36 and 0.31 for the correlation between age and eGFR confirming that significant collinearity was not present.4. It would be far more informative to the reader if Table 1 and 2 presented summary descriptive data from each year of the survey. i.e. Replace Table 1 and 2 with Supplementary Tables 1 and 2. The correlation coefficients and B-values presented in Tables 1 and 2 are not very informative.5. Table 1: r-value for eGFR_{cys} versus survey year = 0. This seems unlikely. Perhaps the authors could confirm that it is correct.
---

	6. Given that the survey interval was 4 or 5 years, the authors should explain how correlation with survey year was conducted. Similarly, how was survey year included in the multivariable model. 7. As shown in Suppl Table 2, the trend in GFR values is not linear and is a bit erratic. For example, the values for 2004 seem unexpectedly low but those for 2014 seem unexpectedly low. Can the authors account for this? 8. The authors report that the prevalence of risk factors increased without presenting any analysis to support this statement. Overall, the approach to statistical analysis seems simplistic. It may be better to employ mixed effects models rather than simple linear regression. In any case, more details regarding the multivariable linear regression analysis should be presented. 9. It would be interesting and clinically relevant to know if the reported change in GFR was also associated with a change in prevalence of CKD. 10. The data are somewhat out of date, as the last year of reporting was 9 years ago. 11. The authors state that similar longitudinal population-based studies have not been performed but multiple such studies have been published e.g.  • Coresh J, Selvin E, Stevens LA, et al. Prevalence of chronic kidney disease in the United States. JAMA 2007;298:2038–47. • Gifford FJ, Methven S, Boag DE, et al. Chronic kidney disease prevalence and secular trends in a UK population: the impact of MDRD and CKD-EPI formulae. QJM 2011;104:1045–53. • Aitken GR, Roderick PJ, Fraser S, et al. Change in prevalence of chronic kidney disease in England over time: comparison of nationally representative cross-sectional surveys from 2003 to 2010. BMJ Open 2014;4:e005480. • Murphy D, McCulloch CE, Lin F, et al. Trends in prevalence of chronic kidney disease in the United States. Ann Intern Med 2016;165:473–81. • Nagata M, Ninomiya T, Doi Y, et al. Trends in the prevalence of chronic kidney disease and its risk factors in a general Japanese population: the Hisayama study. Nephrol Dial Transplant 2010;25:2557–64. • Hallan SI, Vrethuis MA, Romundstad S, et al. Long-term Trends in the prevalence of chronic kidney disease and the influence of cardiovascular risk factors in Norway. Kidney Int 2016;90:665–73. • Juutilainen A, Kastarinen H, Antikainen R, et al. Trends in estimated kidney function: the FINRISK surveys. Eur J Epidemiol 2012;27:305–13. • Lee SW, Kim YC, Oh S-W, et al. Trends in the prevalence of chronic kidney disease, other chronic diseases and health-related behaviors in an adult Korean population: data from the Korean National health and nutrition examination survey (KNHANES). Nephrol Dial Transplant 2011;26:3975–80.
--	---

REVIEWER	Grubb, Anders Lunds Universitet
REVIEW RETURNED	09-Apr-2023

GENERAL COMMENTS	The work by Julia de Man Lapidoth et al describes the development of kidney function in a population in Northern Sweden between 1986 and 2014. One cystatin C-based GFR-estimating equation and two creatinine-based GFR-estimating equations are used for estimation of GFR and the results of all three equations are concordant and indicate that the kidney function has deteriorated during the period studied. The results are interesting, but a few the ambiguities in the manuscript need to be addressed. Pages 2 One author seems to be represented by three different names/shortenings, namely Maria K Svensson, Maria Eriksson Svensson and MKS. Please, clarify! Page 9 The abbreviation “LOQ” should be explained the first time it is used in the manuscript. Page 9 The expression “CRP16vario” is unclear and should be explained or deleted. Page 13 “.....protein increased slightly from 14 to 16 E% from 1990.....” The meaning of E% must be given. Page 15 The sentence: “However, numerous other pollutants that could potentially affect renal function, which are not analyzed in this study.” is unclear and should be corrected. Page 33 The designation “ ?-values” must be specified.
--

REVIEWER	Carsote, Mara Carol Davila University of Medicine and Pharmacy, Endocrinology
REVIEW RETURNED	24-Apr-2023

GENERAL COMMENTS	Dear Authors, The paper is very interesting. The study is well designed and it brings value to our current knowledge. The epidemiologic importance of chronic kidney disease is recognized all over the world. Also, the time dependent changes of this clinical entity might come as a close reflection of cardiovascular diseases and their related negative impact regarding public health. I enjoyed reading the paper and I only have a few minor points. Here are my observations, suggestions or questions.  1. Title – There is a double “in”. I suggest to use “Deterioration of renal function in..” 2. Abstract – Introduction. Second line – “and CKD and..” I suggest to rephrase this, for example, “..is increasing globally, being closely related to cardiovascular diseases”.. 3. Abstract – Study design. I suggest to use “population based study” or an observational, retrospective study, because, otherwise, “repeated cross-sectional studies” means that you repeated the same study. 4. Please explain the abbreviation when first used regardless the Abstract (for example, “CVD”, “EDTA”, “RAAS”).
---

	5. Lines 52-53. "No patient involvement" is not clear since this is an actual study on humans. 6. Measurement – I believe that weight estimation was measured in "0.2" instead of "0,2".. 7. I suggest selecting at least one table or graphic to introduce within main text. Thank you
--	---

VERSION 1 – AUTHOR RESPONSE

Reviewer: 1

Prof. Maarten Taal, Royal Derby Hospital, University of Nottingham

Comments to the Author:

This analysis investigated trends in estimated GFR values in a total of 10,185 participants in the MONICA study in Sweden from 1986-2014. The primary finding was a reduction in mean eGFR over time which remained significant after adjustment for multiple risk factors.

Comments:

1. The relevance of the SCREAM cohort is not clear.

Initially, we would like to thank the reviewer for the insightful remarks and efforts reviewing our manuscript.

We agree and have deleted the reference. We have instead included the CaReMe-study.

2. The introduction discussed the cost of care for people with CKD and ESRD in the US but it may be more relevant to discuss the cost in Sweden.

Thank you for acknowledging this, we agree and have found information about the cost for end-stage renal disease in Sweden. The total cost is approximately 440 million € for dialysis and kidney transplantation. The total cost calculations do not include for instance sick leave or transportation and the total cost is therefore potentially much higher. The paragraph about the cost in the US has been removed from the introduction and we have included the cost estimate for Sweden instead (page 5, line 12-14).

3. Age inclusion changed from 25-64 to 25-74 from 1994. Age was identified as an important determinant of eGFR but not included in the multivariable model because age is a variable in GFR estimating equations. This would only be justified if significant collinearity was observed between eGFR and age. Table 1 reports r-values of 0.36 and 0.31 for the correlation between age and eGFR confirming that significant collinearity was not present.

Thank you for pointing this out. There have been different age groups included in the earlier surveys compared to the latter, which resulted in that the older participants in the latter surveys have been excluded from our study. This has now been clarified in the manuscript (page 9, line 18).

We appreciate your comments and thoughts about our statistics, but we argue that there is a risk of over-adjustment when age is included in the multivariable analysis since age is already a part of the formula for estimating eGFR. However, we have performed an additional multivariate

analysis with age included as an independent variable, showing that survey remained inversely associated with eGFR for both creatinine and cystatin C. Furthermore, high age was significantly associated with lower eGFR.

4. It would be far more informative to the reader if Table 1 and 2 presented summary descriptive data from each year of the survey. i.e. Replace Table 1 and 2 with Supplementary Tables 1 and 2. The correlation coefficients and B-values presented in Tables 1 and 2 are not very informative. Thank you for this comment and yes, this is a mistake from our part, and we thank you for pointing it out. We have now moved Table 1 and 2 to Supplementary tables 1 and 2, and moved Supplementary table 1 and 2 to the manuscript as table 1 and 2 as suggested.

5. Table 1: r-value for eGFR_{cys} versus survey year = 0. This seems unlikely. Perhaps the authors could confirm that it is correct.

We can confirm that this is correct. When performing the Pearson's correlation-analysis the R-value is 0.000 and with a P-value of 0.991. This has been reviewed by a statistician and can be explained by the fact that the linear relationship between eGFR cystatin C and survey year is not strong enough to generate neither a direction nor a valid P-value.

6. Given that the survey interval was 4 or 5 years, the authors should explain how correlation with survey year was conducted. Similarly, how was survey year included in the multivariable model.

Thank you for acknowledging this. This has also been reviewed by a statistician and survey year is included in the linear regression as a continuous variable and as thus, assuming a linear effect. The survey variable is coded as specific year. This might not be apparent in our manuscript and has now been clarified in the statistics section (page 9, line 18).

7. As shown in Suppl Table 2, the trend in GFR values is not linear and is a bit erratic. For example, the values for 2004 seem unexpectedly low but those for 2014 seem unexpectedly low. Can the authors account for this?

Thank you for pointing this out and we have noticed and discussed this as well (and it is also discussed in the manuscript under Strengths and limitations), and the 2004 survey in particular.

A random selection of serum samples from each survey have been re-analysed and the result were similar to the first round.

We have also performed the statistical sensitivity analyses with or without including survey 2004 which gave similar results with declining renal function over time.

8. The authors report that the prevalence of risk factors increased without presenting any analysis to support this statement. Overall, the approach to statistical analysis seems simplistic. It may be better to employ mixed effects models rather than simple linear regression. In any case, more details regarding the multivariable linear regression analysis should be presented.

Thanks. We have carefully checked the manuscript and any statement saying that risk factors were increasing has been corrected.

Regarding performing other statistical analyses instead of linear regression, our study does not include any repeated measurements of the same individuals which makes mixed effect modelling not appropriate. We acknowledge that this might not be evident in our manuscript, and we have now clarified this.

Furthermore, we argue that confidence interval is a better display of the variance and uncertainty in the data, compared to using P-values for presenting descriptive data.

9. It would be interesting and clinically relevant to know if the reported change in GFR was also associated with a change in prevalence of CKD.

This is a very valid point. We have added the percentage of subjects that has an eGFR below 60 ml/min/1.73 m² in Table 2, and the percentage is increasing over time.

10. The data are somewhat out of date, as the last year of reporting was 9 years ago.

Yes, we agree but these data are the available data. New surveys are planned.

11. The authors state that similar longitudinal population-based studies have not been performed but multiple such studies have been published e.g.

- Coresh J, Selvin E, Stevens LA, et al. Prevalence of chronic kidney disease in the United States. *JAMA* 2007;298:2038–47.
- Gifford FJ, Methven S, Boag DE, et al. Chronic kidney disease prevalence and secular trends in a UK population: the impact of MDRD and CKD-EPI formulae. *QJM* 2011;104:1045–53.
- Aitken GR, Roderick PJ, Fraser S, et al. Change in prevalence of chronic kidney disease in England over time: comparison of nationally representative cross-sectional surveys from 2003 to 2010. *BMJ Open* 2014;4:e005480.
- Murphy D, McCulloch CE, Lin F, et al. Trends in prevalence of chronic kidney disease in the United States. *Ann Intern Med* 2016;165:473–81.
- Nagata M, Ninomiya T, Doi Y, et al. Trends in the prevalence of chronic kidney disease and its risk factors in a general Japanese population: the Hisayama study. *Nephrol Dial Transplant* 2010;25:2557–64.
- Hallan SI, Vrethuis MA, Romundstad S, et al. Long-term Trends in the prevalence of chronic kidney disease and the influence of cardiovascular risk factors in Norway. *Kidney Int* 2016;90:665–73.
- Juutilainen A, Kastarinen H, Antikainen R, et al. Trends in estimated kidney function: the FINRISK surveys. *Eur J Epidemiol* 2012;27:305–13.
- Lee SW, Kim YC, Oh S-W, et al. Trends in the prevalence of chronic kidney disease, other chronic diseases and health-related behaviors in an adult Korean population: data from the Korean National health and nutrition examination survey (KNHANES). *Nephrol Dial Transplant* 2011;26:3975–80.

Thank you for enclosing many similar studies and we understand that our wording can be misunderstood.

We still argue that our study never has been performed before using both measurements of creatinine and cystatin C in several cross-sectional observations. None of the studies above are performed in a Swedish cohort and few are performed during this long period of time (30 years). This were not clearly described in the manuscript, and we thank you for pointing this out. We have updated this paragraph (page 13, line 15-21).

Reviewer: 2

Dr. Anders Grubb, Lunds Universitet

Comments to the Author:

The work by Julia de Man Lapidoth et al describes the development of kidney function in a population

in Northern Sweden between 1986 and 2014. One cystatin C-based GFR-estimating equation and two creatinine-based GFR-estimating equations are used for estimation of GFR and the results of all three equations are concordant and indicate that the kidney function has deteriorated during the period studied. The results are interesting, but a few the ambiguities in the manuscript need to be addressed.

Thank you for your wise comments and thoughts concerning our manuscript.

Pages 2

One author seems to be represented by three different names/shortenings, namely Maria K Svensson, Maria Eriksson Svensson and MKS. Please, clarify!

Thank you for acknowledging this and this has now been updated.

Page 8

The abbreviation "LOQ" should be explained the first time it is used in the manuscript.

We agree and have clarified this in the manuscript (page 8, line 15-16).

Page 9

The expression "CRP16vario" is unclear and should be explained or deleted.

We agree and this is now deleted.

Page 13

".....protein increased slightly from 14 to 16 E% from 1990....." The meaning of E% must be given.

This has been clarified in the manuscript. E has been removed as suggested this E does not add any meaningful information (page 14, line 10-11).

Page 15

The sentence: "However, numerous other pollutants that could potentially affect renal function, which are not analyzed in this study." is unclear and should be corrected.

Thank you for pointing this out, the sentence has been adjusted and clarified in the manuscript (page 14, line 17-18).

Page 33

The designation " ?-values" must be specified.

We thank you for noticing this and it has been changed in the manuscript (page 30, line 17).

Reviewer: 3

Dr. Mara Carsote, Carol Davila University of Medicine and Pharmacy

Comments to the Author:

Dear Authors,

The paper is very interesting. The study is well designed and it brings value to our current knowledge. The epidemiologic importance of chronic kidney disease is recognized all over the world. Also, the time dependent changes of this clinical entity might come as a close reflection of cardiovascular diseases and their related negative impact regarding public health.

I enjoyed reading the paper and I only have a few minor points.

Here are my observations, suggestions or questions.

1. Title – There is a double “in”. I suggest to use “Deterioration of renal function in..”

Thank you for your kind words, comments, and thoughts on our manuscript. We have changed the title, according to the editor’s suggestion (see above).

2. Abstract – Introduction. Second line – “and CKD and..” I suggest to rephrase this, for example, “..is increasing globally, being closely related to cardiovascular diseases”..

We agree and has been rephrased according to your suggestion.

3. Abstract – Study design. I suggest to use “population based study” or an observational, retrospective study, because, otherwise, “repeated cross-sectional studies” means that you repeated the same study.

Thank you for noticing this, and we have made changes in line with your recommendation (page 2, line 7).

4. Please explain the abbreviation when first used regardless the Abstract (for example, “CVD”, “EDTA”, “RAAS”).

Thank you for this comment and this has been clarified in the manuscript.

5. Lines 52-53. “No patient involvement” is not clear since this is an actual study on humans.

We agree that this is misleading and has been rephrased (page 6, line line 19-20).

6. Measurement – I believe that weight estimation was measured in “0.2” instead of “0,2”.

This has been changed as suggested.

7. I suggest selecting at least one table or graphic to introduce within main text.

We have carefully considered this suggestion but found that te manuscript became overloaded, and that Figure 1 gives a quick overview of the main results. If the editor prefers so, we can try to produce another graphic.

VERSION 2 – REVIEW

REVIEWER	Grubb, Anders Lunds Universitet
REVIEW RETURNED	19-Jun-2023
GENERAL COMMENTS	The authors have used the important suggestions by the reviewers in the revised manuscript